



# A Large-Eddy Simulation Approach for Wind Turbine Wakes and its Verification with Wind Tunnel Measurements

Jiangang Wang, Chengyu Wang, Filippo Campagnolo, and Carlo L. Bottasso

Wind Energy Institute, Technische Universität München, Garching bei München, Germany

*Correspondence to:* C.L. Bottasso (carlo.bottasso@tum.de)

**Abstract.**

This paper first describes a large-eddy simulation approach, and then verifies it with respect to single-turbine wind tunnel experiments. Various aspects of the numerical approach are considered, to try to reduce its need for tuning, improve its accuracy and limit its computational cost. Simulation

results are compared to measurements, including rotor and wake quantities. The study includes normal operating conditions, as well as wake manipulation by derating, yaw misalignment and cyclic pitching of the blades. Results indicate a good overall matching of simulations with experiments. Low turbulence test cases appear to be more challenging than moderate and high turbulence ones, due to the need for denser grids to limit numerical diffusion and accurately resolve tip-shed vortices

in the near wake region.

## 1   Introduction

Wind plants are collections of wind turbines, often operating in close proximity of one another. Several complex phenomena take place within a wind farm. First, there is an interaction between the atmospheric boundary layer and the whole wind farm, caused by the smaller scale interaction between the atmospheric flow and each individual wind turbine. Second, within the power plant it-

self, there is an interaction among upstream and downstream wind turbines through their wakes. In turn, the wake themselves interact with the atmospheric flow and other wakes, interactions that play a central role in determining the overall behavior of the plant. Wakes produced by upstream wind turbines may have a profound influence on the performance of downstream operating machines. In

fact, waked turbines experience lower power output and increased loading, compared to clean isolated conditions. A thorough understanding of these complex phenomena is clearly indispensable for optimizing the layout and operation of wind plants. However, even an optimal layout will still incur in wake interactions, at least in some wind and environmental conditions. To mitigate these effects, a number of control strategies are currently being investigated to optimize the operation of



wind power plants, including power derating, wake deflection and enhanced wake recovery (Knudsen et al., 2015; Fleming et al., 2014).

The current research in this field is very active, covering a broad spectrum that ranges from high-fidelity numerical simulations to reduced order models, from scaled experiments in the wind tunnel to direct measurements in the field, all the way to control methods and various supporting tech-

nologies. Among the many studies reported in the literature, meteorological and performance data collected at the Horns Rev and Middelgrunden offshore wind farms can be mentioned as examples of full scale field data that have been systematically investigated (Hansen et al., 2012; Barthelmie et al., 2007). Moreover, scaled wind farm experiments were conducted in wind tunnels to study wake deficit and its impact on downstream wind turbines (Bartl et al., 2012; Chamorro and Porté-

Agel, 2009; Medici and Alfredsson, 2006). These test campaigns have been actively used to validate several engineering and CFD wake models, in terms of power capture, velocity profiles and higher order flow quantities (Barthelmie et al., 2006; Gaumond et al., 2014; Kennedy et al., 2011; Porté-Agel et al., 2011). Wake models can be classified on the basis of their complexity and fidelity to reality. The steady-state kinematic wake model of Jensen (1983) was among the first proposed ana-

lytical formulations, later extended by Jiménez et al. (2010) to cover the case of yaw misalignment. Larsen et al. (2007) derived a more sophisticated dynamic wake meandering model. Higher-fidelity models have been developed by using computational fluid dynamics (CFD). For example, Carcangiu (2008) used the Reynolds-averaged Navier Stokes (RANS) equations to simulate the near wake behavior, while Stovall et al. (2010) used the same turbulence model to simulate wind turbine cluster

conditions and compare RANS to the higher-fidelity large-eddy simulation (LES) approach. Results indicated that RANS is not sufficiently accurate, as it typically overestimates diffusion.

With the significant increase in computational performance of the recent years (thanks to advancements in hardware, software and algorithms), LES has gained an increasing adoption by the wind farm research community (Churchfield et al., 2012; Porté-Agel et al., 2011; Calaf et al., 2010). In

fact, LES has the ability to better resolve the relevant flow features, leading to an improved insight on flow characteristics within a wind farm. In addition, several researchers (Jiménez et al., 2010; Fleming et al., 2014; Gebraad et al., 2016) have been using LES to investigate wake control strategies.

Although LES is an approach based on first principles, it is still not completely tuning-free. For

example, when used in conjunction with an actuator line method (ALM) to represent wind turbine blades, there is a need for properly tuning the procedure used for mapping lifting line aerodynamic forces onto the volumetric grid (Martinez et al., 2012). In addition to several algorithmic details, other important characteristics of the simulation are represented by the grid and features of the model, including the presence (or absence) of nacelle and tower. In general, most of the published

research focuses on the use of CFD to study wake behavior and control strategies, but pay relatively less attention to the problem of ensuring the fidelity of such simulations to reality. In fact, a com-



prehensive validation of LES methods for wind turbine wakes is still missing. This is clearly not due to a lack of attention to this problem, but rather to a lack of comprehensive high-quality data sets. Unfortunately, experiments in the field are not without hurdles: in fact, wind conditions cannot

be controlled, and measurements at full scale are not always possible nor complete. In this sense, testing at scale in a wind tunnel is gaining attention as a means to perform experiments with a much more precise knowledge and control of the testing conditions.

As a contribution towards a better understanding of the capabilities and limits of LES for modeling wind turbine wakes, this paper applies a recently developed computational framework to the

simulation of scaled wind turbines. These models were operated in a large boundary layer wind tunnel in a variety of conditions. A complete digital model of the experiments is realized within the LES framework, which also includes a model of the wind tunnel and of the passive generation of sheared and turbulent flows.

The present LES framework is characterized by some distinguishing features. First, the tuning-

free immersed boundary (IB) method of Jasak and Rigler (2014) is used to model the effects caused by nacelle and tower. Second, the integral velocity sampling method (Churchfield et al., 2017) is employed, which reduces the sensitivity of the results —and especially of power— to the mapping of aerodynamic forces onto the fluid flow. Third, an ad-hoc developed approach is used for tuning the airfoil polars. In fact, given the small scale of the experimental models, their blades operate at

low Reynolds numbers and are therefore designed using special low-Reynolds airfoils. Clearly, the accuracy of the airfoil polars plays an important role in the accuracy of the overall LES simulation. Rotational augmentation, manufacturing imperfections and other effects may influence the behavior of the blade airfoils and alter it with respect to their nominal characteristics, which are typically obtained in 2D dedicated wind tunnel tests. To address this issue, airfoil polars are tuned here by

means of a specific identification method (Bottasso et al., 2014), which makes use of dedicated experimental measurements conducted with the scaled turbine (i.e., not with the single airfoils, but with the rotor on which the airfoils are used). Indeed, the airfoil Reynolds varies depending on the operating condition of the turbine. By accounting for the effects of Reynolds on the airfoil polars, which are particularly relevant at the low Reynolds numbers at which the scaled models operate, a

better accuracy in the results can be achieved.

The problem of computational cost is addressed in a companion paper (Wang et al., 2018b), where a scale adaptive simulation (SAS) approach is used to model the unresolved scales, resulting in a LES-like behavior at a cost similar to RANS, with a roughly similar accuracy.

The paper is organized according to the following plan. The numerical method is described in

Sect. 2. The computational setup is reported in Sect. 3, where a precursor simulation —mimicking the process that takes place in the wind tunnel— is used for the passive generation of the turbulent flow, whose resulting outflow is used as inlet for subsequent wind turbine wake simulations (called successor simulations). The experimental setup is presented in Sect. 4, including a short description



of the wind tunnel, of the scaled wind turbine model and of the measurement equipment. Results are
discussed in Sect. 5. At first, an isolated flow-aligned wind turbine is considered, and the LES frame-
work is tuned to match experimental measurements obtained in this baseline case. Next, the three
wake manipulation strategies of derating, yaw misalignment and cyclic pitch control are considered.
Here again low-turbulence experimental results are compared with simulations, without any addi-
tional tuning with respect to the parameters chosen in the baseline case. Finally, a moderate turbulent
condition is considered, again without any additional tuning. Conclusions are drawn in Sect. 6.

## 2   Numerical simulation model

The present LES framework is developed within SOWFA (Churchfield and Lee, 2012; Fleming et al.,
2013), a simulation tool based on a standard incompressible solver in the OpenFOAM repository.
The rotor is modelled in terms of actuator lines, by direct coupling with the aeroservoelastic simula-
tor FAST (Jonkman and Buhl Jr, 2005). The integral approach of Churchfield et al. (2017) is used to
compute the flow conditions at each station along an actuator line, and to project the calculated aero-
dynamic forces back onto the fluid domain using a single Gaussian width value. Aerodynamic forces
at each station are computed by interpolating pre-computed lift and drag aerodynamic coefficients,
which are stored in look-up tables parameterized in terms of angle of attack and Reynolds number.
The IB formulation of Jasak and Rigler (2014) is used to model the wind turbine nacelle and tower. A
blended algorithm is implemented to control numerical dispersion and diffusion: the Gamma scheme
(Jasak et al., 1999) is used in the near wake region, while central differencing is used in the far wake.
The second order implicit backward scheme is used for time marching. Depending on the problem,
the wind turbine model is either controlled in closed-loop by a pitch and torque controller, based on
the implementation described in Bottasso et al. (2014), or simply by using experimentally measured
values of pitch and rotor speed.

### 2.1   Sub-grid scale model

LES requires a model to represent the unclosed Reynolds stress tensor $\tau_{ij}^r$ at sub-grid scales. The
approach adopted in this paper uses a functional artificial eddy-viscosity model, which is formulated
as

$$\mu_t = (C_s h)^2 |\mathbf{S}|, \tag{1}$$

where $C_s$ is the Smagorinsky constant, $h$ is the grid size and $\mathbf{S}$ the strain-rate tensor. Two well-known
methods are investigated to determine the value of $C_s$, namely the Constant Smagorinsky model
(Deardorff, 1970) (termed here CS model), and the Lagrangian averaging dynamic Smagorinsky
model (Meneveau et al., 1996) (named LDS model in the following). The CS model simply uses
a specified time-invariant $C_s$ value throughout the whole simulation domain, while the LDS model
solves two additional transport equations to dynamically determine a local $C_s$ value. Therefore, the



LDS approach uses a temporally and spatially adaptive eddy viscosity model, while the CS method uses a constant $C_s$ value. In principle, the LDS model should be capable of achieving more accurate

results than the CS model, at the cost of a higher computational cost. However, results of extensive numerical experiments indicate that, for the present application, the performance of the LDS model is very similar to the CS model, as shown later on in this work.

### 2.2 Immersed boundary method

The IB formulation of Jasak and Rigler (2014); Lai and Peskin (2000); Mittal and Iaccarino (2005)

is used to model the wind turbine nacelle and tower, whose effects on the flow proved to be quite significant —at least in the near wake region— and should therefore not be neglected (Wang et al., 2017b). The IB method is employed to avoid the use of surface conforming meshes to represent the shape of such bodies (Mittal and Iaccarino, 2005). The present IB approach, based on a discrete forcing method, uses a direct imposition of the boundary conditions (Uhlmann, 2005), this way

preserving the sharpness of the body shape. Boundary conditions and wall models can be directly imposed on the IB surfaces with this approach, yielding good solution quality for high Reynolds viscous flows (Bandringa, 2010). Details of the formulation are described in Wang et al. (2017b).

### 2.3 Numerical discretization and solution of the resulting linear systems

The ALM introduces a source (body-force) term in the governing equations. This produces a certain

amount of numerical dispersion if central differences are used for discretizing the convection term. The projected body forces induce local velocity increments, which translate into high Péclet numbers (Pe) (Moukalled et al., 2016). In fact, the Péclet number provides for a measure of numerical dispersion and writes

$$\text{Pe} = \frac{\rho u h}{\Gamma},\tag{2}$$

where $\rho$ is the air density, $u$ is the flow speed, and $\Gamma$ the diffusion coefficient. Indeed, Moukalled et al. (2016) considered the numerical solution of an inviscid steady state one-dimensional momentum equation obtained by discretizing the convection term with second-order central differences. Results indicate that the numerical solution departs from the analytical one for Péclet numbers larger than 2.

Increases in Pe caused by ALM create numerical dispersion, but are typically not high enough

to cause simulations to crash. On the other hand, the IB-modeled nacelle and tower introduce sharp edges and irregular surfaces that, generating high local Péclet numbers, cause large numerical dispersion and may eventually result in the failure of the simulation. The Péclet number can be reduced by increasing the grid resolution, which is however undesirable due to its resulting impact on the computational cost. In addition to velocity increases, IB-induced local flow misalignments with respect

to the grid can also destabilize a simulation (Holzmann, 2016; Moukalled et al., 2016).





To limit numerical dispersion and diffusion, the deferred-correction Gamma-bounded high-resolution interpolation method is used here in conjunction with the IB formulation (Jasak et al., 1999). The Gamma scheme is parameterized in terms of $\beta_m$, a tunable constant that allows one to control the level of upwinding. In general, a larger value of $\beta_m$ implies a lower dispersion and a higher diffusion (i.e. more upwinding), and vice versa. The value $\beta_m$=0.45 is employed in the near wake region to stabilize the simulation, since actuator line body forces and immersed boundary possibly generate numerical dispersion, and $\beta_m$=0.05 is used in the far wake to minimize numerical diffusion while retaining a minimum amount of necessary upwinding.

The linear algebraic solvers and grid quality also play important roles in determining the computational efficiency of the overall LES framework (Greenshields, 2015). Simulations are conducted using high-quality grids, with about 99% of cubic cells for the wind turbine successor cases, and small time-steps (Courant number = 0.3). Moreover, significant flow separation or adverse pressure gradients are typically absent in the present application. Because of this, the use of advanced pre-conditioners and low-quality-mesh-correctors (Greenshields, 2015) is not essential, with beneficial effects in terms of computational efficiency. Table 1 shows the linear solvers used for the precursor and the wind turbine/wake simulations. The precursor problem has slightly less regular grids, because of the need to mesh the large turbulence generators (termed spires) placed at the tunnel inlet, which requires a slightly different setup of the linear solvers. The conjugate gradient (CG) method is employed to solve the equations with symmetric matrices, associated with pressure $p$, while a bi-CG is used for the asymmetric matrices associated with $\tilde{\mathbf{u}}$ and $T$. The geometric-algebraic multi-grid method is used as pre-conditioner for pressure, while the diagonal incomplete-LU factorization is used for asymmetric matrices. Gauss-Seidel is used as a smoother for the pre-conditioning of $p$ for the turbine simulation, while the diagonal incomplete Cholesky factorization Gauss-Seidel is used for the precursor case on account of its lower grid quality, which increases the computational cost by about 5%.

The PISO time marching algorithm allows for recursively solving (or correcting) the pressure flux equation to account for non-orthogonal grid elements (Greenshields, 2015). The number of iterations is fixed a priori and set equal to 1 and 0 for the precursor and successor simulations, respectively. Indeed, given the good quality of the grid in the latter case, non-orthogonal corrections are not indispensable, and their elimination lowers the computational cost by about 10%.

## 2.4 Multi-airfoil table identification

As mentioned earlier, the ALM is used to represent the effects of wind turbine blades on the fluid flow. The ALM works by providing a description of the blades in terms of their chord and twist distributions, as well as in terms of sectional lift and drag coefficients for a number of airfoils located at specific stations along the blade span. At each instant of time, the angle of attack and relative flow speed at a generic air-station are computed by combining the blade overall motion with the local



| Type | Precursor simulation | Wind turbine simulation |
|---|---|---|
| $p$ solver | CG | CG |
| $p$ preconditioner | GAMG | GAMG |
| $p$ smoother | DIC-GS | GS |
| No. of $p$ corrector steps | 3 | 3 |
| $\tilde{\mathbf{u}}$ solver | bi-CG | bi-CG |
| $\tilde{\mathbf{u}}$ preconditioner | DILU | DILU |
| No. of NOC steps | 1 | 0 |

**Table 1.** Linear algebraic solvers used for the precursor and the wind turbine/wake simulations (CG = conjugate gradient; GAMG = geometric-algebraic multi-grid; DIC = diagonal incomplete Cholesky; GS = Gauss-Seidel; DILU = diagonal incomplete LU factorization; NOC = non-orthogonal corrector).

flow velocity, which is sampled in an appropriate spatial neighborhood in the fluid domain. Based on angle of attack, Reynolds number and blade geometry at the given spatial location along the blade, the aerodynamic coefficients are interpolated from the stored look-up tables, based on which

local instantaneous values of the sectional aerodynamic force components can be readily computed. Finally, these force components are mapped back onto a suitable neighboring fluid volume, providing for a body-force source term in the governing fluid motion equations.

Clearly, the accuracy of the sectional aerodynamic coefficients is a crucial ingredient of the whole procedure. A method to tune the aerodynamic polars of lifting lines was described in Bottasso et al.

(2014). In a nutshell, the method works by first measuring thrust and torque on a rotor at a number of different operating conditions that cover the angles of attack and Reynolds numbers of interest. Next, these values are used to update some given baseline polars by using a maximum-likelihood criterion.

In this work, the polar identification method of Bottasso et al. (2014) is used. Nominal values

of both the lift and drag coefficients $C_k$ (where $k = L$ or $k = D$ for lift and drag, respectively) are corrected as

$$C_k(\eta, \alpha, \mathrm{Re}) = C_k^0(\eta, \alpha, \mathrm{Re}) + \Delta_k(\eta, \alpha, \mathrm{Re}), \qquad (3)$$

where $\eta \in [0, 1]$ is a span-wise location, $\alpha$ the angle of attack, Re the Reynold number, $C_k^0$ the nominal coefficient value, and $\Delta_k$ is the unknown correction. This latter term is expressed by a

linear interpolation as

$$\Delta_k(\eta, \alpha, \mathrm{Re}) = \mathbf{n}^T(\eta, \alpha, \mathrm{Re})\mathbf{p}_k, \qquad (4)$$

where $\mathbf{p}_k$ is the vector of unknown nodal values and $\mathbf{n}(\eta, \alpha, \mathrm{Re})$ is the vector of assumed multi-linear shape functions. To improve the well-posedness of the problem, the polar correction terms



are transformed using a singular-value decomposition, which ensures the actual observability of the
tuned parameters. By this method, the corrections to the baseline lift and drag characteristics of
the airfoils are recast in terms of a new set of statistically independent parameters. By analysing
their associated singular values, one can retain in the identification only those parameters that are
observable with a desired level of confidence (Bottasso et al., 2014).

The unknown correction terms are computed by maximizing the likelihood function of a sample of
$N$ available experimental observations. This amounts to first minimizing the following cost function

$$J = \frac{1}{2} \sum_i^N \mathbf{r}_i^T \mathbf{R}^{-1} \mathbf{r}_i, \tag{5}$$

where $\mathbf{r}$ is the discrepancy between power and thrust coefficients computed by a blade element
momentum model, as implemented in the `WT-Perf` code (Buhl, 2009), and the corresponding ex-
perimentally measured quantities. The optimization is performed for a fixed covariance $\mathbf{R}$, by using
the gradient-based sequential quadratic programming approach. Next, the covariance is updated as
$\mathbf{R} = 1/N \sum_i^N \mathbf{r}_i \mathbf{r}_i^T$, and the optimization is repeated. Iterations between minimization and covari-
ance update are continued until convergence (Bottasso et al., 2014).

More than one hundred operating points were measured experimentally. The operating conditions
were determined in order to cover a desired range of angles of attack and Reynolds numbers, and
were obtained by operating the scaled wind turbine model at different tip-speed-ratios (TSRs) and
blade pitch angles. Experiments were then grouped in terms of average blade Reynolds number, and
for each group a separate identification was performed, yielding a calibrated version of the polars at
that specific Reynolds.

## 245    3   Computational setup

The LES-ALM numerical model is used to create a complete digital copy of the experimental setup.
Experiments were conducted in the 36 m × 13.84 m × 3.84 m boundary layer test section of the wind
tunnel at Politecnico di Milano (Bottasso et al., 2014; Zasso et al., 2005), which is a closed-return
facility powered by 16 fans for a total of 1.5 MW.

### 250    3.1   Precursor simulation

#### 3.1.1   Computational mesh

A first simulation is used to generate the turbulent inflow (precursor) used as inlet for successive
wind turbine/wake (successor) simulations. The layout of the partially overlapped precursor and
successor domains is shown in Fig. 1. The precursor domain has a size of 30 m × 6.92 m × 3.84 m.
The reduced width of the domain with respect to the actual tunnel size is chosen to reduce the com-
putational cost. The turbulent inflow for the successor simulation is sampled 19.2 m downstream of




the precursor inlet, as highlighted in the figure. The simulation mimics the passive turbulence gener-
ating system adopted in the wind tunnel where experiments were conducted (Zasso et al., 2005). A
structured body-conforming mesh discretizes the volume around the turbulence-generating spires at
the wind tunnel inlet, using a purely hexahedral O-grid. The average stretching ratio for the volume
mesh is 1.25, while the maximum skewness is equal to 2.7, which however does not compromise
the simulation stability. Mesh quality is limited by the sharp edges and abrupt surface changes of
the spire geometry. The mesh resolution is designed not to fully resolve the boundary layer, and
the average $y^+$ is equal to 50. A wall-modeled simulation significantly reduces computational costs
compared with a fully resolved one. In fact, the precursor mesh used in this work contains 59 mil-
lion cells, while one or two orders of magnitude more cells would be necessary to achieve $y^+ = 1$
with a structured grid as the present one, with consequently extremely high computational costs. In
addition, as discussed later, results show that the wall-modeled boundary layer approach is capable
of providing good agreement with the experimental wind tunnel measurements.

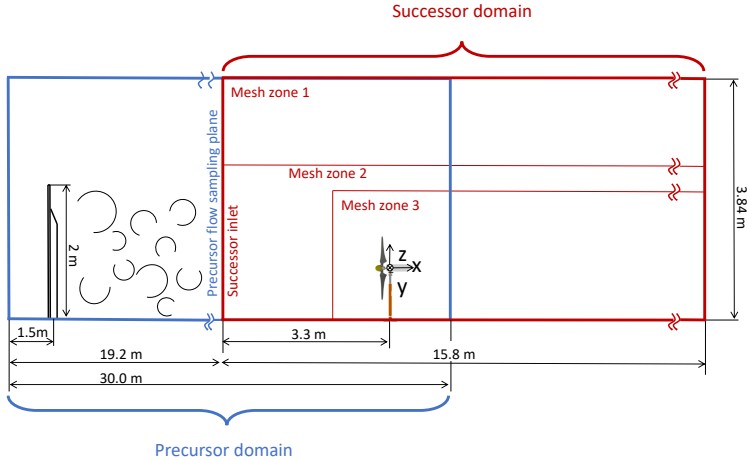

**Figure 1.** Layout of the partially overlapped precursor and successor computational domains.

**3.1.2   Boundary conditions**

Dirichlet-type non-slip conditions are used for the resolved velocity vector $\tilde{\mathbf{u}}$ on the tunnel side walls
and the spire surfaces. Neumann-type conditions are imposed for pressure $p$ on the same boundary
surfaces, while Dirichlet-type wall conditions are employed for temperature $T$, which is assumed to
be the same on all surfaces. Regarding the sub-grid scale model, Dirichlet-type surface conditions
are used for the eddy viscosity $\mu_t$ on the ceiling, using a fixed value equal to $1 \times 10^{-5}$ m$^2$/s on
account of the negligible turbulence of the viscous sublayer; a small positive non-zero value is used,





because $\mu_t$ is evaluated at cell centroids and not on the wall surface. A wall model is imposed on the other surfaces including spires, left/right walls and floor to adjust wall shear stresses.

The inflow speed at the inlet equals 4.7 m/s, as measured in the wind tunnel, and the maximum

Courant number is limited to one. The constant Smagorinsky sub-grid scale model is used, with $C_s$ set to 0.13. In order to reach steady-state conditions, the simulation requires about 15 s of physical time. After achieving a steady mean speed, the precursor flow is collected at a sampling plane about 3D in front of the turbines and stored, to be used as input for subsequent successor simulations.

### 3.1.3    Verification of the precursor

Figure 2 shows the normalized time-averaged streamwise velocity $\langle u_x \rangle$ and turbulence intensity $\sigma/\langle u_x \rangle$ profiles measured 20.85 m downstream of the tunnel inlet, which corresponds to 1.5D upstream of the wind turbine rotor. A reference frame is located at the hub, as shown in Fig. 1 on the right. The two horizontal and vertical velocity profiles are in good agreement with the experimental data. The average velocity error $\langle \Delta u_x \rangle$ is around 1-2% for both profiles. Turbulence intensity also

shows a reasonable agreement, with an average error of 7% and 5% for the horizontal and vertical profiles, respectively. The experimental results for $\sigma/\langle u_x \rangle$ along the horizontal profile show an unexpected discontinuity, not observed in the simulations, which might be due to the effect of the traversing system used for holding and positioning the hot wire probes.

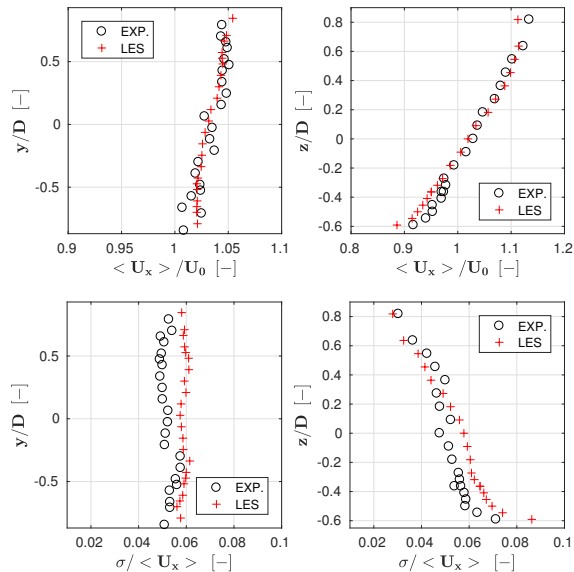

**Figure 2.** Normalized time-averaged streamwise velocity $\langle u_x \rangle$ (top row) and turbulence intensity $\sigma/\langle u_x \rangle$ (bottom row), 1.5D downstream of the rotor. Left column: hub-height horizontal profile; right column: hub-centered vertical profile. Red + symbols: numerical results; black ∘ symbols: experimental measurements.



Figure 3 shows the experimental and simulated turbulent kinetic energy spectrum $E(f)$ and auto-
correlation $r(\tau)$ at hub height, 1.5D upstream of the rotor. The LES-computed spectrum appears to
be in good agreement with the experimental one. The autocorrelation is computed as:

$$r^j(\tau) = \left\langle (u_x^j(t) - \langle u_x^j \rangle)(u_x^j(t+\tau) - \langle u_x^j \rangle) \right\rangle, \tag{6}$$

where $u_x^j$ is the streamwise component of the velocity at spatial point $j$. The integral time scale
(O'neill et al., 2004), defined as

$$T_\tau^j = \int_0^\infty \frac{r^j(\tau)}{\left\langle u_x^{j,2} \right\rangle} \mathrm{d}\tau, \tag{7}$$

is found to be 0.139 s and 0.143 s for experiment and simulation, respectively. These results indicate
a good overall agreement between simulation and experiment even at small scales, with a consequent
correct estimation of flow mixture, wake recovery and other relevant features of the flow.

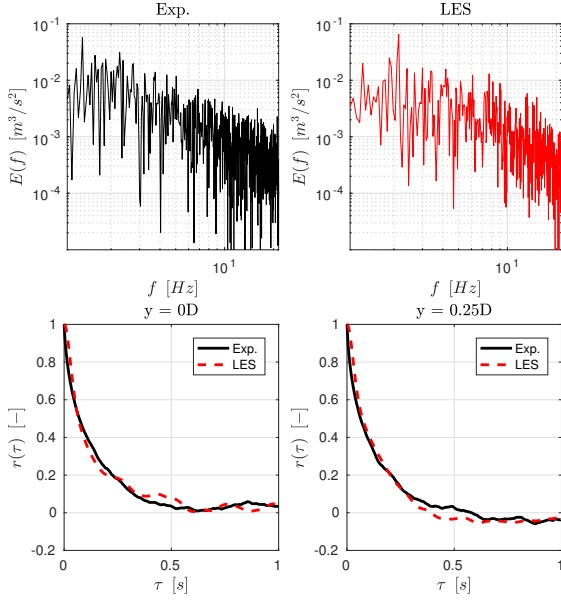

**Figure 3.** Turbulent kinetic energy spectrum $E(f)$ at hub height 1.5D upstream of the rotor, for the experiment
(top-left, black line) and simulation (top-right, red line). Autocorrelation $r(\tau)$ at hub height 1.5D upstream of
the rotor (bottom-left) and 0.25D to its left, looking downstream (bottom-right).





### 3.2 Successor simulation

#### 3.2.1 Computational mesh

The computational setup for the wind turbine/wake simulation follows Wang et al. (2017a). The domain layout is shown in Fig. 1. The domain width is reduced to 3.9D, which is 3.4 times less than the actual test section width, so as to minimize the computational cost without affecting the results due to wall blockage. Notice that the precursor width is about twice the width of the successor

domain, simply because the same precursor is also used for non-aligned multi-turbine configurations (not discussed here) that, having a larger frontal area, require a larger inflow. The mesh uses three zones of increasing density. Zone 1 is the base mesh, with cubic cells of 0.08 m in size, while zones 2 and 3 have cubic cells of 0.04 m and 0.01 m, respectively. Less than 1% of the total mesh is composed of polyhedral cells, while all others are cubic.

#### 3.2.2 Boundary conditions

Two different flow conditions are considered in the present study. In the first case, the flow velocity is obtained from a LiDAR-scanned low turbulence ($< 2\%$) inflow condition (van Dooren et al., 2017). Measurements also accounts for a slight non-uniformity of the flow within the wind tunnel (Wang et al., 2017a). In the second case, as previously explained, the output of the passively-generated

turbulent precursor simulation was instead used as inlet for the successor simulation.

The treatment of the domain walls is as follows. Dirichlet-type non-slip wall conditions for $\tilde{\mathbf{u}}$ are used for the tunnel ceiling and floor. Neumann-type conditions for $p$ and $T$ and mixed type conditions for $\tilde{\mathbf{u}}$ are used for the side walls, enforcing a null component of the velocity normal to the side surfaces to ensure mass conservation. The eddy viscosity $\mu_t$ is set with Neumann conditions on

the left/right tunnel walls. For ceiling and floor, $\mu_t$ is set with Dirichlet conditions to the fixed value $1 \times 10^{-5}$ m$^2$/s in the low turbulence case, while a wall model is used for the moderate turbulence condition.

Dirichlet-type non-slip wall conditions are used for the IB-modeled nacelle and tower in the low turbulence case. In fact, in these cases a laminar boundary layer (or, at least, a not fully-developed

turbulent boundary layer) is expected to extend over the entire IB surface due to the steadiness of the incoming flow. Despite the maximum $y^+$ being equal to 50 on the IB surfaces, a wall function can no be used here, as it could properly model only a fully developed turbulent boundary layer. Due to the coarse grid, an overestimation of the boundary layer thickness on the IB-modeled bodies is expected, which in turn will lead to an overestimation of the blockage induced by the turbine nacelle

and tower.

Slip wall IB surface conditions are used for the moderate turbulence case, in order to mitigate numerical stability issues. Although this neglects the boundary layer induced blockage and turbulence, results indicate a negligible impact on the downstream wake profile. This is probably explained by





the background turbulence that, by enhancing mixing, diffuses the signature of tower and nacelle on
the downstream flow.

## 4    Experimental setup

### 4.1    Scaled wind turbine model

Tests were performed with the G1 scaled wind turbine model, whose rotor diameter and optimal
TSR are equal to 1.1 m and 8.25, respectively. The model, already used within other research projects
(Campagnolo et al., 2016c, a, b), is designed to have realistic wake characteristics, with shape, deficit
and recovery that are in good accordance with those of full-scale machines. The model features active
individual pitch, torque and yaw control that, together with a comprehensive onboard sensorization
(including measures of shaft and tower loads), enables the testing of turbine and farm-level control
strategies.

### 4.2    Wake measurement

The flow within the wind tunnel was measured with hot-wire probes or stereo PIV. The latter tech-
nique was used to measure the flow characteristics in the near (0.56D) and far (6D) wake regions.
The measurement planes cover a significant fraction of the wind turbine wake. In order to achieve a
higher spatial resolution of the velocity field, the measurement area was divided into several windows
with small overlaps between them. A rapid scanning of the entire measurement area was achieved
by the use of an automated traversing system, moving both the laser and the cameras. The measuring
windows were divided into 32×32 pixel interpolation areas, which resulted in an approximatively
15 mm spatial resolution. For each measuring window, 200 pairs of images were acquired (per
camera) without phase lock, resulting in time-averaged flow field measurements. Additional details
concerning the PIV instrumentation are given in Campanardi et al. (2017).

## 5    Result and analysis

### 5.1    Baseline simulation and parameter tuning

The baseline simulation represents an isolated flow-aligned wind turbine. The machine is operated
in a low turbulence flow, with a rotor-averaged inflow velocity equal to 5.9 m/s, which is slightly
lower than the G1 rated speed (6.0 m/s).

This first case is used to determine the optimal values of the Smagorinsky constant $C_s$ and of the
Gamma scheme parameter $\beta_m$. The same tuned parameters are used for all other simulations in the
rest of this work. This first test case is also used to verify the effects of the Gaussian width $\epsilon$, which
is used to project aerodynamic forces from the lifting lines onto the computational grid. In fact, it





was observed that this projection may have a significant effect on the results, including rotor power and thrust. In principle, $\epsilon$ should be set equal to 2-3 times the cell size, i.e. $2h \leq \epsilon \leq 3h$ (Martinez et al., 2012). It was found that the dependency of the rotor aerodynamic power on $\epsilon$ is significantly reduced if the integral velocity sampling approach is used (Churchfield et al., 2017). For instance, if $\epsilon$ increases by 30%, power will increase by 13% if the traditional point-wise velocity sampling

approach is used, but only by 5% when using the integral velocity sampling method. In fact, in the point-wise approach a variation of $\epsilon$ reshapes the Gaussian curve, in turn changing the peak value and eventually affecting the calculated aerodynamic power, while the integral approach uses a weighted average that mitigates the reshaping effect (Churchfield et al., 2017).

Using a simple trial and error approach, the three parameters $\epsilon$, $C_s$ and $\beta_m$ (in the near wake)

were set to 0.025, 0.13 and 0.45, respectively. Given the low turbulence of the present case, the experimentally measured rotor speed was very nearly constant, and its average value was used in the simulation.

The rotor integral quantities of power and thrust are compared first, by time-averaging over 10 s. The wind turbine power was found to be equal to 45.79 W in the experiment, and equal to 45.45 W

for LES, showing a good agreement between these two values. A slightly larger discrepancy was obtained for the thrust, which was found to be 15.18 N and 16.05 N for experiment and simulation, respectively. This may be explained by the fact that thrust is directly measured at the shaft in the numerical simulation, while it is reconstructed from the tower based fore-aft bending moment in the experiment. This requires estimating the contribution of nacelle and tower, which is done by a

dedicated experiment performed on the wind turbine without the blades. As a result, this indirect calculation of the experimental thrust is affected by approximations, and it cannot be regarded as accurate as the measurement of rotor torque (and hence of power).

Next, the characteristics of the wake are compared between PIV measurements and CS LES simulation. Figure 4 shows streamwise velocity contours on a plane 0.56D downstream of the rotor.

Measurements are missing from two areas left and right of the rotor disk where, due to the close proximity of the measuring plane with the wind turbine, part of the nacelle (which is of a white color) was in the background, leading to a wrong correlation between the PIV images. Apart from the two missing spots, the LES contours are similar to the PIV ones, both in terms of wake width and deficit. The wake deficit for LES is on average 1.3% higher than the experiment.

The figure also shows that the simulation overestimates the local wake deficit behind the nacelle and tower, as a results of the enhanced blockage effect mentioned in §3.2. Indeed, the current mesh resolution (high $y^+$) implies a thicker boundary layer, which in turn produces a higher blockage with a consequent larger flow separation, tower shedding and induced turbulence. This problem could be mitigated by a suitable refinement of the mesh near the IB, which however would come at the price

of a significant increase in the computational cost.



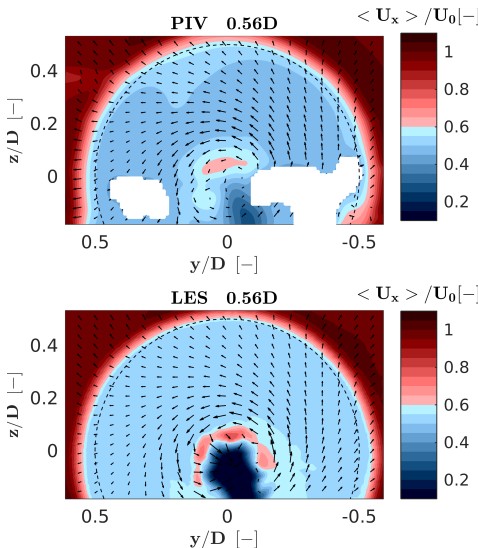

**Figure 4.** Streamwise velocity contours for the CS LES model and PIV experimental measurements, on a plane 0.56D downstream of the rotor. Black arrows indicate the cross-wind velocity component at a number of sampling points.

Next, hot-wire probe measurements are used to compare wake profiles at 3D, 4D, 7D and 8D downstream positions. Figure 5 shows horizontal (top row) and vertical (central row) profiles of the normalized time-averaged velocity, as well as horizontal profiles of turbulence intensity (bottom row). The plots report results for the CS model, the LDS model, and experimental measurements.
The CS case includes two sets of results, one obtained including the effects of nacelle and tower in the model by the IB approach, and one obtained neglecting these two components. Comparing these two curves with the experimental results clearly indicates that the near wake profile is more accurately represented when nacelle and tower are included in the model, as already noted by other authors (Santoni et al., 2017). This may be particularly true for the present scaled wind turbine, for
which these two components are relatively bigger than in full scale machines. Indeed, the sum of the frontal area of the nacelle and of the portion of the tower located within the rotor swept area $A$ is $0.037A$, while it is $0.023A$ for the NREL 5 MW wind turbine (Jonkman et al., 2009). Although this parameter is larger for the G1, it is expected that the effects of nacelle and tower on wake evolution might not be negligible even for typical multi-MW wind turbines (Wang et al., 2017b). All other
simulations reported in this work were performed including nacelle and tower in the model.

Both CS and LDS show a good agreement with the experimental curves. Indeed, the temporally and spatially averaged streamwise velocity difference $\langle \Delta u_x \rangle = (\langle u_{x,\text{LDS}} \rangle - \langle u_{x,\text{CS}} \rangle)/\langle u_{x,\text{CS}} \rangle$ between the CS and LDS models is consistently less than 1% at all downstream distances. Results





indicate that the LDS model does not provide significantly more accurate results than the CS one,
while at the same time it requires a 20% larger computational effort caused by the solution of its
two extra transport equations. Moreover, turbulence intensity plots seem to indicate a slightly better
match of CS to the experiments than LDS. Based on these results, all other simulations in the present
paper were based on the CS model.

The rotor-averaged streamwise velocity difference between simulation (with nacelle and tower)
and experiment $\langle \Delta u_x \rangle = (\langle u_{x,\text{sim}} \rangle - \langle u_{x,\text{exp}} \rangle)/\langle u_{x,\text{exp}} \rangle$ is equal to -2.7%, -1.6% and -1.3% at 3D,
4D, and 8D, respectively. The root mean square (RMS) error can be used to quantify the spatial fit
between simulations and experiments, and it writes

$$\text{RMS}(\cdot) = \sqrt{\frac{1}{N} \sum_{j=1}^{N} \left( \left\langle (\cdot)_{\text{sim}}^j \right\rangle - \left\langle (\cdot)_{\text{exp}}^j \right\rangle \right)^2}, \tag{8}$$

where $\left\langle (\cdot)^j \right\rangle$ is a generic time-averaged quantity at a given spatial point $j$. At the various downstream
distances, $\text{RMS}(u_x)$ equals 0.34 m/s, 0.33 m/s and 0.15 m/s, respectively. As expected, the matching
of simulation with experimental measurements improves moving downstream. Indeed, if rotor thrust
is well predicted, flow mixture is properly resolved and numerical diffusion is suitably controlled,
then the simulation results in a fully developed wake that correlates well with the experiment. The
far wake profile can be approximated by the single Gaussian distribution used in some engineering
wake models (Larsen et al., 2007; Renkema, 2007).

LES underestimates the rotor-averaged turbulence intensity $\sigma/\langle u_x \rangle$ by 23%, 12% and 12% at 3D,
4D, and 8D, respectively, while the rotor-averaged root mean square error $\text{RMS}(\sigma/\langle u_x \rangle)$ is 0.04,
0.02 and 0.02 at these same positions. The turbulence intensity profiles of Fig. 5 clearly show that
matching is not as good as in the case of the streamwise velocity, especially in the near wake region
where tip vortices are not resolved enough and tower shedding is overpredicted. Here again, the
problem could mitigated with a finer grid, which however would lead to increased computational
costs.

Comparing the turbulence intensity results with and without nacelle and tower shows that there is
an increased turbulence in the wake of the former case, which causes an earlier vortex breakdown and
produces higher turbulence intensity at the far wake. In turn, this generates a faster wake recovery, as
shown in the speed deficit plots. Here again, this confirms the need for including nacelle and tower
in the simulation.

### 5.2 Low turbulence inflow simulation

In this section, the characteristics of the LES framework are assessed with reference to three wake
control strategies, namely power derating (or axial induction control), wake steering by yaw mis-
alignment and wake enhanced recovery by cyclic pitch control. The flow conditions and setup of the
simulations are the same described earlier in the baseline case.



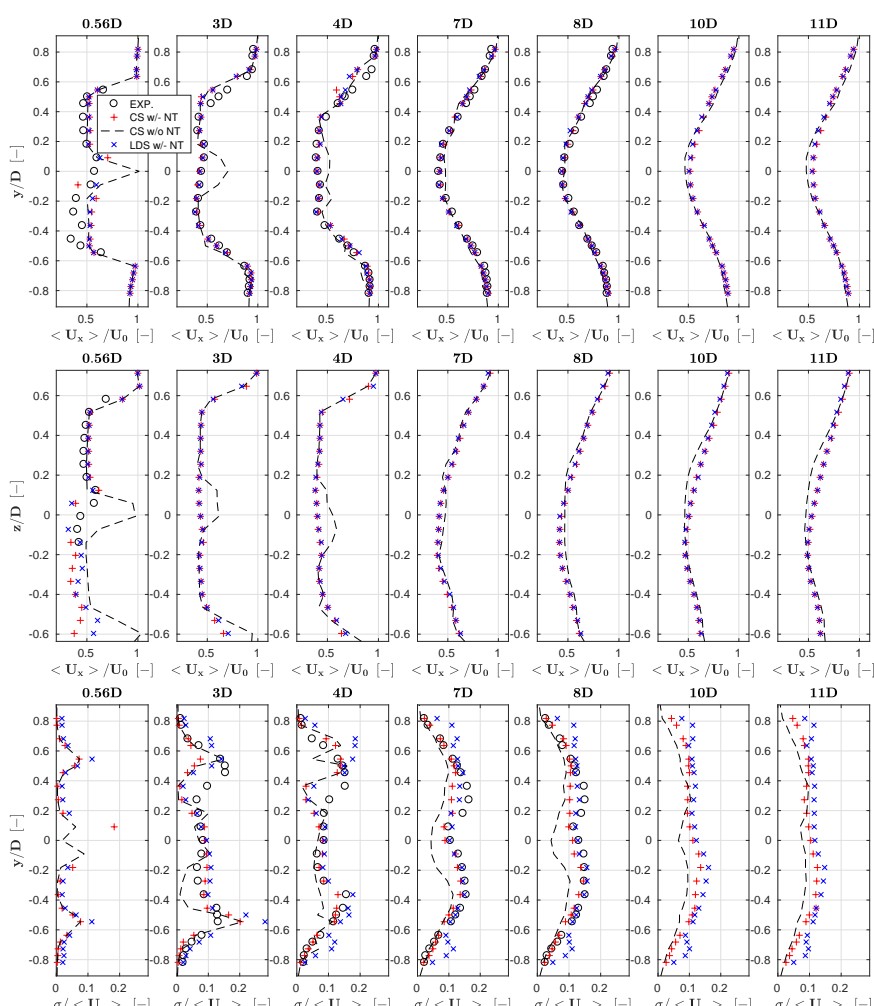

**Figure 5.** Profiles of normalized time-averaged streamwise velocity $\langle u_x \rangle / U_0$ along hub height horizontal lines (top row), and along vertical lines through the hub (middle row). Turbulence intensity $\sigma / \langle u_x \rangle$ along hub height horizontal lines (bottom row). Red $+$ symbols: CS model with nacelle and tower; black dashed line: CS model without nacelle and tower; blue $\times$ symbols: LDS model with nacelle and tower; black $\circ$ symbols: experimental results.





### 5.2.1 Power derating

Power derating was accomplished in the experiment by providing the turbine power controller with
modified values of the rotor speed and torque. Specifically, for a power partialization factor $p_f$, the
reference rotor speed is modified as $\sqrt[3]{p_f}\Omega$, while the torque as $\sqrt[3]{p_f^2}Q$. This corresponds to having
set the rated wind speed to the value $\sqrt[3]{p_f}U_\infty$; since this is lower than the current wind speed $U_\infty$,
the machine is now effectively operating in the full power region. Therefore, the collective blade
pitch controller automatically adjusts the pitch setting to track the new reference rotor speed.

The resulting pitch and rotor speed changes modify the angle of attack and Reynolds number at
the blade sections. Therefore tests that include power derating are useful for evaluating the quality
of the identified multi-airfoil tables. Indeed, to accurately estimate rotor power and thrust, the lifting
line airfoil polars need to match the aerodynamic characteristics of the corresponding blade sections,
in order to generate and project the proper body forces onto the fluid domain.

Simulations are conducted by prescribing the rotor speed and blade pitch measured in the experiment. Four power settings are considered, namely 100%, 97.5%, 95% and 92.5% of rated power.
Figure 6 shows wake velocity profiles measured at hub-height at a 4D downstream position. For
all cases, rotor-averaged speed error $\langle\Delta u_x\rangle$ and RMS($u_x$) are about 1% and 0.25 m/s, respectively.
A quite satisfactory agreement between the simulation and experimental results can be noticed,
although partialization seems to have only a modest effect on wake profile. Turbulence intensity
profiles are not presented here, since the quality of the comparison is very similar to the one of the
baseline case.

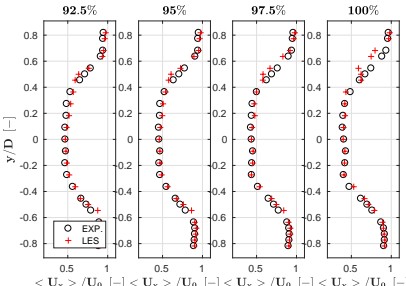

**Figure 6.** Normalized time-averaged streamwise velocity $\langle u_x\rangle/U_0$ profiles at 100%, 97.5%, 95% and 92.5%
power settings (from right to left), measured at hub height and 4D downstream of the rotor. Red + symbols:
LES; black ○ symbols: experimental results.

However, the situation is less satisfactory for rotor power and thrust, as indeed shown in Table 2.
Results indicate that power is particularly off, while thrust is affected by somewhat smaller errors.
This might indicate a possible discrepancy in the drag of the airfoil polars. To improve this aspect of





the model, a new calibration of the polars is being developed that includes also a possible spanwise variability of the Reynolds number.

|  |  | 100% | 97.5% | 95% | 92.5% |
|---|---|---|---|---|---|
| Power | Exp. $[W]$ | 45.79 | 44.36 | 43.20 | 42.11 |
|  | Sim. $[W]$ | 45.45 | 42.28 | 39.72 | 37.33 |
|  | $\Delta P \%$ | -0.74 | -4.69 | -8.06 | -11.35 |
| Thrust | Exp. $[N]$ | 15.18 | 14.24 | 13.62 | 13.10 |
|  | Sim. $[N]$ | 16.05 | 14.57 | 13.56 | 12.70 |
|  | $\Delta T \%$ | 5.73 | 2.32 | -0.44 | -3.05 |

**Table 2.** Power and thrust at 100%, 97.5%, 95% and 92.5% power settings.

### 5.2.2 Wake steering by yaw misalignment

Next, the LES model is verified in yaw misalignment conditions, which are relevant to wake de-
flection control. Hub-height wake profiles measured in low turbulence conditions are used for the comparison, for yaw misalignment angles of $\pm 5$ deg, $\pm 10$ deg and $\pm 20$ deg.

Simulated and measured longitudinal speed profiles are presented at a downstream distance of 4D in Fig. 7. Similar results were obtained at other distances, but are not reported for space limitations. The maximum rotor-averaged difference $\langle \Delta u_x \rangle$ between simulation and experiment is 4.1% and
corresponds to the 20 deg case, while the maximum $\mathrm{RMS}(u_x)$ is 0.35 m/s at -10 deg. The average $\langle \Delta u_x \rangle$ and $\mathrm{RMS}(u_x)$ over the six yaw cases are equal to 1.6% and 0.29 m/s, respectively. The results indicate a good agreement between simulation and measurement, both in terms of wake deficit and pattern. Notice, however, that the 1.6% average speed error would correspond to a 4.8% power error for a second wind turbine operating in full wake shading at this downstream difference, a value that
is small but not completely negligible.

### 5.2.3 Enhanced wake recovery by cyclic pitch control

A third wake control strategy in the same low turbulence conditions is considered, where the rotor blades are cyclicly pitched. The effect of cyclic pitching is that of changing the angle of attack of the blade sections cyclically over one rotor revolution. In turn, this results in an azimuthal change of the
out of plane forces generated by the section, which has then the effect of correspondingly modifying the local induced velocity. A simple analytical model of the effects of cyclic pitching was developed in Wang et al. (2016). The analysis showed that, as already noticed by other authors (Fleming et al., 2014), CyPC has some effect on the speed of recovery of the wake, but results only in a very modest deflection of its path. In fact, wake deflection by yawing is driven by the tilting of the rotor thrust, which results in a significant lateral force being applied onto the flow. On the other hand, CyPC




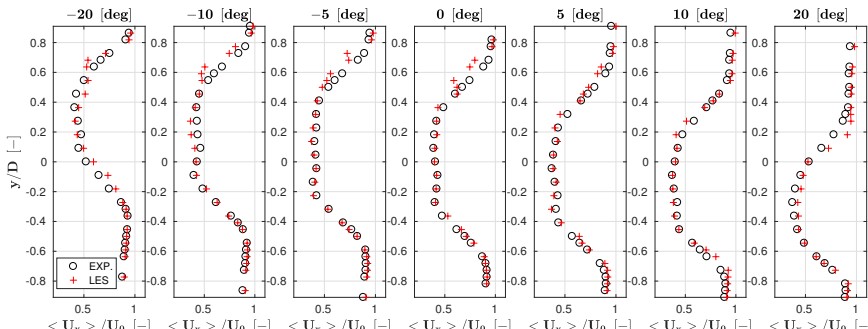

**Figure 7.** Normalized time-averaged streamwise velocity profiles at hub height for different yaw misalignments, 4D downstream of the rotor. Red + symbols: LES; black ○ symbols: experimental results.

does modify the induced velocity, but only generates negligible lateral forces. In addition, it was observed that CyPC also results in large moments being generated in the rotor fixed frame, which further questions the practical applicability of this wake manipulation strategy. Nonetheless, CyPC is considered here to further verify the characteristics of the LES framework in operating conditions that differ significantly from the ones of the previous test cases.

Each blade is pitched according to $\theta_i = \theta_0 + \theta_c \cos(\psi_i + \gamma)$, where $\theta_0$ is the collective pitch constant, $\theta_c$ the 1P pitch amplitude, $\psi_i$ the blade azimuth angle (clockwise looking downstream, and null when the blade is pointing vertically up), and $\gamma$ is the phase angle (with the same origin and positive sense as $\psi$). The CyPC parameters were set as $\theta_0$=0 deg, $\theta_c$=5.3 deg, and $\gamma$=270 deg.

Given the effects of CyPC on the induced velocity and on the near wake behavior, a more complete analysis can be performed by using the PIV measurements than considering the simple hub-height line scans obtained by hot wire probes. Figure 8 reports, at left, the streamwise velocity just behind the rotor ($x/D$=0.56), which is a distance where few results have been previously reported. The images show that the use of CyPC has a strong effect on the wake structure, leading to a marked unsymmetrical shape. Indeed, the phase angle $\gamma$=270 deg implies that blades have maximum pitch, and hence produce the minimal rotor-plane-normal force, in the left part of the rotor —as shown in the figure—, which in turn exhibits the lowest induction and highest resulting longitudinal flow speed. A comparison between experimental and numerical results shows that there is, in general, a good qualitative agreement and that the main distortion effects caused by CyPC are reasonably captured. The rotor-average error $\langle \Delta u_x \rangle$ between simulation and measurement is 2.69%, while RMS$(u_x)$ is 0.79 m/s.

The discrepancy between simulation and experiment is two times larger than in the baseline case. One possible reason for this is that unsteady aerodynamic effects of the airfoils (including dynamic stall) are neglected. This could be improved by using unsteady aerodynamics models in the lifting line, including for example a Theodorsen correction and a dynamic stall model. Although the



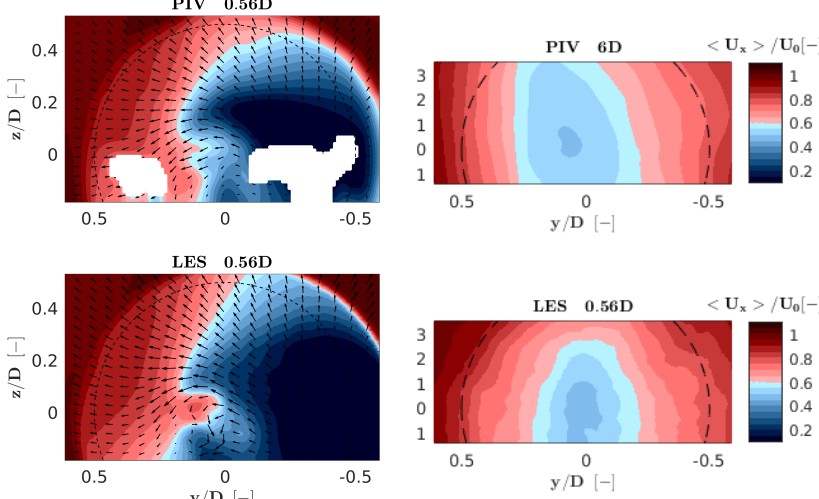

**Figure 8.** Streamwise velocity contour plots for the PIV measurements (top row) and LES model (bottom row), measured 0.56D (left) and 6D (right) downstream of the rotor. Black arrows indicate the cross-wind velocity component at a number of sampling points.

Beddoes-Leishman approach (Moriarty and Hansen, 2005) is implemented in `FAST` and therefore could be readily used in the present LES framework, the model requires the definition of several airfoil-dependent parameters, which would need to be specifically calibrated for the low-Reynolds airfoils used on the G1 scaled wind turbine.

The comparison of LES and experiment in the far wake (6D) is slightly better, as it can be observed in the right part of the same figure. The wake recovery is reasonably good in terms of flow speed, although the slight tilting towards the right shown by the PIV measurements is not apparent in the LES results. Lastly, it should be remarked that CyPC leads to a faster recovery of the wake than in the baseline case, as already noticed by Wang et al. (2016). In principle, this could be of interest for

wind farm control, although, as previusly mentioned, the large resulting loads exerted on the rotor probably limit the practical applicability of this control concept.

### 5.3   Moderate turbulence inflow simulation

Next, a turbulent case is considered, where a flow characterized by a 6% hub-height turbulence intensity is generated by the precursor simulation described in §3.1. The wind turbine model is

aligned with the streamwise flow direction and the hub-height wind speed is equal to 4.76 m/s. The simulated wind turbine operates in two different modes, namely with a fixed rotating speed of 720 RPM and blade pitch angle of 1.4 deg (which are the values measured on the scaled model in the



experiment) or with a controller in the loop (Bottasso et al., 2014). Since the machine operates in the partial load region, the blade pitch setting is constant and torque control is based on a pre-computed look-up table.

The aerodynamic power output, averaged over a 60 s time window, is equal to 31.0 W for the experiment, and to 30.5 W and 31.2 W for the prescribed speed and closed-loop torque simulations, respectively. In this latter case, the average rotor speed was only 2.2% higher than the one measured on the wind turbine, which clearly indicates a good overall match of the numerical model with the experiment. On the other hand, the power standard deviation was 0.2 W, 0.6 W and 0.3 W, respectively for the experiment, prescribed speed and closed-loop simulations. Clearly, prescribing a constant speed to the rotor in the numerical simulation induces significant torque oscillations, because the rotor cannot adjust to the turbulent flow fluctuations. When loads are of interest, it is therefore essential to use a closed-loop controller also in the simulation. However, in this case the simulation might drift away from the operating condition realized in the experiment, if the numerical model has a significant mismatch with respect to reality. Apparently, this is not the case here, and the numerical model seems to be well in line with the experimental one.

Figure 9 shows the normalized time-averaged velocity and turbulence intensity profiles for the LES model and experiment, at distances of -1.5D, 1.4D, 1.7D, 2D, 3D, 4D, 6D and 9D from the rotor. The position at -1.5D is outside of the induction zone, and the flow can be regarded as being the undisturbed free stream. The LES curves show, in general, a good agreement with the experimental ones. Only the case of the closed-loop regulation is reported here, as results are nearly identical to the prescribed-speed case. The rotor-averaged simulation error $\langle \Delta u_x \rangle$ is less than 1% on average across all distances. From the near (1.4D) to the far (9D) wake regions, the root mean square error $\mathrm{RMS}(u_x)$ gradually reduces from 0.2 m/s to 0.1 m/s. The comparisons all indicate that the LES results are in good agreement with the experimental ones.

Contrary to the baseline low-turbulence simulation, the two turbulence intensity peaks induced by the blade tip vortices are well predicted in this case. To explain this phenomenon, we report in Fig. 10 for the low (left) and moderate (right) turbulence cases the instantaneous streamwise speed component $u_x/U_0$, the vorticity $\langle \nabla \times \mathbf{u} \rangle$ and the turbulence intensity $\sigma/\langle u_x \rangle$, on a horizontal plane at hub height. As previously observed, the turbulent structures induced by nacelle and tower are different for the two cases, on account of the different boundary conditions on their surfaces.

Vorticity shed by the tips in the near wake is quite similar for the low and moderate turbulence cases. In fact, the actuator line model of the blade tips produces a similar amount of velocity shear for both the low and moderate turbulence conditions. Turbulence intensity is, on the other hand, very different in the blade tip region for these two different ambient turbulence cases. In fact, the higher background turbulence of the turbulent inflow case triggers the instability of the tip vortical structures (Sørensen, 2011), which rapidly break down. The contour plots of the turbulent simulation clearly shows that, starting from 0.1D downstream, the tip vortices generate significant turbulence intensity,



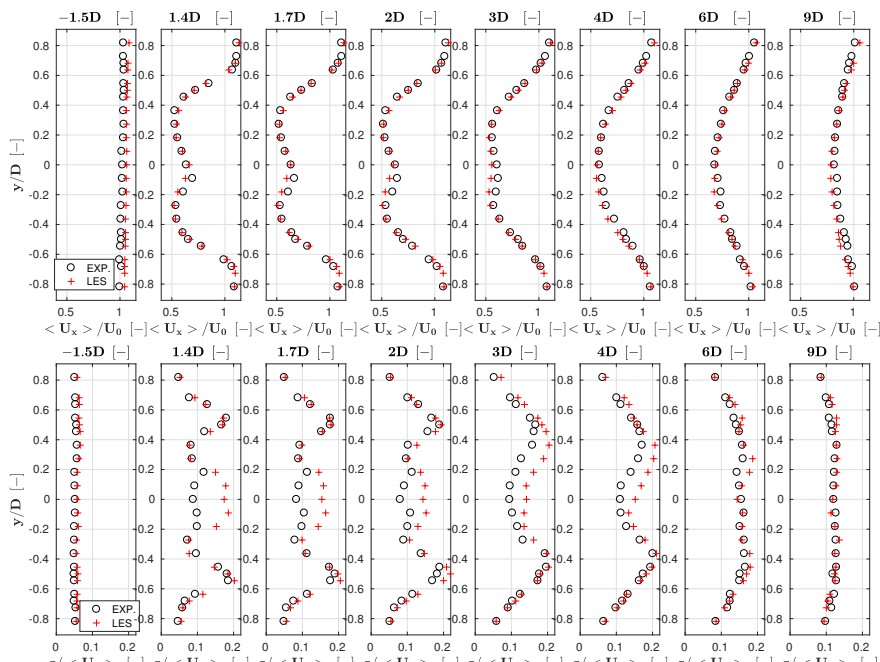

**Figure 9.** Normalized time-averaged streamwise velocity $\langle u_x \rangle / U_0$ (top row) and turbulence intensity $\sigma / \langle u_x \rangle$ (bottom row) profiles at hub height. Red + symbols: LES; black ○ symbols: experimental results.

while vorticity quickly diminishes from 2D downstream, signalling that the coherent tip vortices have broken down into smaller and less coherent structures. Quite differently, the low turbulence case shows a persistent modest turbulence intensity and high vorticity up to about 4D downstream of the rotor. In this case, capturing the right amount of speed fluctuations —which are here mostly caused by the tip vortices, in contrast to the other case that is predominantly dominated by turbulent

fluctuations— probably requires a denser grid than the one used here, and this explains the poor match with the experiments in this case in the near-wake tip region. Apparently, the same grid is however capable of representing well the simpler turbulent case. An analysis of tip vortex breakdown is reported in Troldborg et al. (2015) using a blade conforming approach, which therefore uses very significantly denser grids than in the present case.

**6   Conclusions**

This paper has described a LES approach for the simulation of wind turbine wakes. The simulation model was used to develop a complete digital copy of scaled wind tunnel experiments. The proposed approach is here characterized with respect to wind tunnel measurements of one single wind turbine,





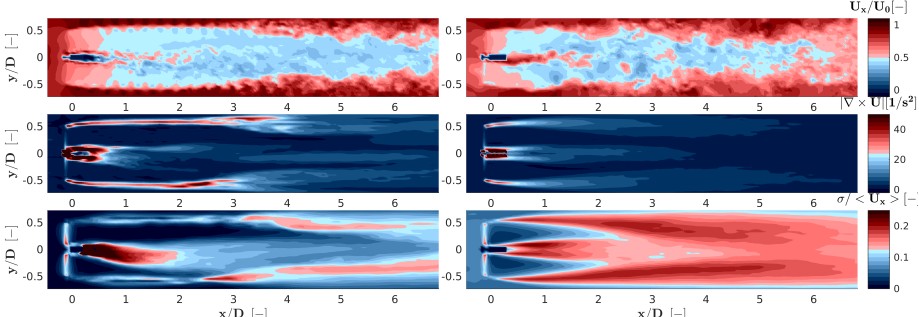

**Figure 10.** Instantaneous streamwise speed component $u_x/U_0$ (top row), vorticity $\langle \nabla \times \mathbf{u} \rangle$ (central row), and turbulence intensity $\sigma/\langle u_x \rangle$ (bottom row). At left, low turbulence case; at right: moderate turbulence case.

while multiple machines and wake interactions have been studied in Wang et al. (2017b, 2018a) and in other forthcoming papers.

The integral velocity sampling method is used for the ALM, while nacelle and tower are accounted for by an IB approach. An identification method is used for the calibration of airfoil polars, accounting for the low and variable Reynolds at which the scaled wind turbines operate. For increased realism of the results, the simulation model can optionally operate with the same pitch-torque controller in the loop used on board the experimental model. A low Courant number and high-quality meshes are used to speed up simulations, which in fact are run without sophisticated preconditioners or non-orthogonal grid correctors. The simulation of one experiment is divided in two steps: a precursor first generates a sheared and turbulent flow, whose outflow becomes the inflow for a successor that simulates the wind turbine and its wake.

A low turbulence normal-operation problem is considered first, showing that simulations are in good agreement with experiments, both in terms of rotor quantities (thrust and power) and wake behavior. Next, the three wake control strategies of power derating, wake steering by yaw misalignment and wake enhanced recovery by cyclic pitch control are studied. Results show a good agreement of simulations with experiments for yaw misalignment, but are less satisfactory for derating, probably on account of inaccuracies in the airfoil drag. The wake turbulence intensity shows some discrepancies, which were here attributed to a lack of refinement of the grid that in turn affects the breakdown of the near wake vortical structures. Slightly less accurate results are obtained for cyclic pitching, possibly due to un-modeled unsteady airfoil aerodynamics.

The paper continues by considering a moderately turbulent wind. The characteristics of the simulated turbulent flow are in good agreement with measurements. The average streamwise velocity is within 1% of the experiments, the average turbulence intensity within 5-7%, while the turbulent kinetic energy spectrum and integral time scale also exhibit a good matching. The wake characteristics are in very good agreement with the experiments, since tip vortices break down earlier than in



the low turbulence condition, relaxing a bit the need for very dense grids in the near wake region.
The use of a controller in the loop leads to a more realistic response of the model turbine to the turbulent flow, which is important in case the load response of the machine is of interest. Remarkably, the model in the loop also operates at essentially the same rotor speed as the experiment, which demonstrates the overall fidelity of the digital model to the experimental one.

Results shown in this work indicate that the present LES-ALM approach is a viable way of sim-
630 ulating scaled wind tunnel experiments. Results are however not perfect, and areas of improvement include a more sophisticated and accurate calibration of the airfoil polars, the inclusion of airfoil unsteady aerodynamic effects (which however also call for the calibration of these models with dedicated data sets), and a more efficient refinement of the grid where necessary by the use of unstructured meshing and adaption techniques.

These encouraging results motivate and justify the application of the present simulation framework to the analysis of clusters of wake-interacting wind turbines, for which we have gathered an ample collection of data sets in multiple operating conditions. Hopefully, this will lead to a better understanding of wake behavior, which is of crucial importance for the design and operation of wind turbines and wind power plants. The final validation of the present and similar simulation approaches
can undoubtedly benefit from the use of scaled wind tunnel experiments, as attempted in this work, as an intermediate step towards their application to the full scale case.

*Acknowledgements.* This work has been supported in part by the CL-WINDCON project, which receives funding from the European Union Horizon 2020 research and innovation program under grant agreement No. 727477. The first author was supported by the Chinese Scholarship Council. All tests were performed
at the wind tunnel of the Politecnico di Milano, with the support of Prof. A. Croce, Dr. Zanotti, Mr. G. Campanardi and Mr. D. Grassi. The authors wish to thank Mr. E.M. Nanos of the Technische Universität München and Dr. V. Petrović now at the University of Oldenburg for their contribution to the experimental work. The authors also express their appreciation to the Leibniz Supercomputing Centre (LRZ) for providing access and computing time on the SuperMUC Petascale System.

**Nomenclature**

| | |
|---|---|
| $A$ | Rotor swept area |
| $D$ | Rotor diameter |
| $C_L$ | Lift coefficient |
| $C_D$ | Drag coefficient |
| $C_k^0$ | Nominal coefficient |
| $C_s$ | Smagorinsky constant |
| $E(f)$ | Turbulent kinetic energy spectrum |
| $J$ | Cost function |





|      |                        |                                         |
|------|------------------------|-----------------------------------------|
|      | $N$                    | Number of available experimental observations |
| 660  | $Q$                    | Rotor torque                            |
|      | $\mathbf{R}$           | Covariance matrix                       |
|      | $\mathbf{S}$           | Strain-rate tensor                      |
|      | $T$                    | Temperature                             |
|      | $T_r$                  | Integral time scale                     |
| 665  | $U_\infty$             | Free-stream wind speed                  |
|      | $h$                    | Grid size                               |
|      | $k$                    | Turbulent kinetic energy                |
|      | $p$                    | Pressure                                |
|      | $p_f$                  | Power partialization factor             |
| 670  | $r(\tau)$              | Auto-correlation                        |
|      | $u_i$                  | Velocity component                      |
|      | $y^+$                  | Dimensionless wall distance             |
|      |                        |                                         |
|      | $\alpha$               | Angle of attack                         |
| 675  | $\beta_m$              | Tunable constant for Gamma scheme       |
|      | $\gamma$               | Phase angle                             |
|      | $\epsilon$             | Gaussian width                          |
|      | $\eta$                 | Span-wise location                      |
|      | $\theta$               | Blade pitch angle                       |
| 680  | $\nu_t$                | Eddy viscosity                          |
|      | $\psi$                 | Blade azimuthal angle                   |
|      | $\rho$                 | Density                                 |
|      | $\sigma/\langle u_x \rangle$ | Turbulence intensity              |
|      | $\tau_{ij}^r$          | Reynolds stress tensor                  |
| 685  | $\tau$                 | Time shift for autocorrelation analysis |
|      |                        |                                         |
|      | $\Gamma$               | Diffusion coefficient                   |
|      | $\Omega$               | Rotor speed                             |
|      |                        |                                         |
| 690  | $\Delta\cdot$          | Correction or difference                |
|      | $\langle\cdot\rangle$  | Averaged quantity                       |
|      | $\tilde{(\cdot)}$      | Resolved quantity                       |
|      |                        |                                         |
| 695  | ALM                    | Actuator line method                    |





| | | |
|---|---|---|
| | bi-CG | Bi-conjugate gradient |
| | CG | Conjugate gradient |
| | CFD | Computational fluid dynamics |
| | CS | Constant Smagorinsky |
| 700 | CyPC | Cyclic pitch control |
| | DIC-GS | Gauss-Seidel smoothing with diagonal incomplete Cholesky factorization |
| | DILU | Diagonal incomplete-LU factorization |
| | GAMG | Geometric-algebraic multi-grid |
| | IB | Immersed boundary |
| 705 | LES | Large-eddy simulation |
| | LiDAR | Light Detection and Ranging |
| | LDS | Lagrangian averaging dynamic Smagorinsky |
| | NOC | Non-orthogonal corrector |
| | PISO | Pressure-Implicit with Splitting of Operators |
| 710 | PIV | Particle Image Velocimetry |
| | RANS | Reynolds-averaged Navier Stokes |
| | RMS | Root mean square |
| | TSR | Tip speed ratio |
| | | |
| 715 | Pe | Péclet numbers |
| | Re | Reynolds number |





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
