# Peer review of "Wake behavior and control: comparison of LES simulations and wind tunnel measurements"

_Wind Energy Science, 2018_

## Referee Comment (RC1) · Anonymous Referee #1 · 25 Jul 2018

This manuscript deals with LES simulations of a down-scaled wind turbine model, and their assessment against wind tunnel data. My main comment is that this manuscript reads more as a technical report rather than a scientific paper. Majority of the text is devoted to technical details of the LES that often are not novel and already well established in the scientific community.

In my opinion, if the focus of the paper is proposing new capabilities for an LES solver for wind turbine wake simulations, then these potential novelties should be highlighted already from the abstract. Furthermore, for each control capability, assessment against wind tunnel data for one or few conditions might not be enough to assess the accuracy

of the model. Maybe, the work can be extended by detecting flow features that were not easily and well detected through the wind tunnel experiments and now better singled out through the LES simulations.

I suggest to significantly revise the focus and scope of the manuscript.

————————————————

---

## Referee Comment (RC2) · Anonymous Referee #2 · 31 Aug 2018

Summary The paper compares wind tunnel experiments of a wind turbine with large-eddy simulations using an actuator line model. The results are interesting, however, the reviewer suggests a major revision before publication in WES. There are various improvements that need to be made before the paper is ready for publication.

The paper is too long and there are many unnecessary details about the simulations that do not need to be there. Also, the authors missed many references in the introduction, although some of them were cited later during the discussion of the results. The authors should collapse many of the sections and instead of explaining all the details about linear solver, etc, they should just cite the appropriate references. Many of the

details are not relevant to the findings in the paper.

The authors conclude that the main differences in power and thrust between experiment and simulations come from the blade polars. However, it looks more like the differences are coming from turbine parametrization, such as epsilon, integral sampling, etc.

Specific comments:

In the introduction, when talking about ALM, the authors are missing some references. Please cite the original reference of the ALM (Sorensen and Shen 2002). There are good guidelines in the literature for choosing the appropriate tuning parameters for the ALM, but they require very fine grid resolutions (Jha et al JSEE 2014, Martinez et al WE 2017, etc). Also, the effect of nacelle and tower has been studied by many (Churchfield et al AIAA 2015, Santoni et al. WE 2017, Stevens et al, RE 2018, Yang and Sotiropoulos WE 2018, etc).

In the section Sub-grid scale model, there are no references to the work of others. The effect of SGS model and Smagorinsky coefficient on wind turbine wakes has been studied before (Sarlak et al RE 2015, Martinez et al. JRSE 2018, etc).

The section "Numerical discretization and solution of the resulting linear systems" seems to give too much information and not all of it is necessary. This section should be cleaned up.

In the section "Computational mesh" there is a complicated explanation about wall model and y plus. I believe this explanation is not needed and the authors can just mention that a wall model is used.

Figure 2: Why is the velocity not symmetric about y=0?

"Baseline simulation and parameter tuning" It is not clear how the optimal parameters were chosen. Do these values agree with the optimal values found in the literature? Please add this to the discussion.

Figure 5: Please move the legend outside of the figures. It would be best to just show the spanwise profiles in this figure (y/D). If it is needed, another figure showing the wall normal direction (z/D) can be added.

Figure 6: Can you make the plots wider, such that differences are shown better?

"Power derating" The authors conclude that the difference in polars is causing the differences in power. It seems unlikely that the polars would change much over the small Reynolds number variations seen by the blade. It seems that these differences are more due to the turbine parametrizations (epsilon, integral sampling, etc).

---

## Author Comment (AC1) · 12 Dec 2018

**Reply to Reviewers**

We thank the reviewers for their detailed analysis and constructive inputs. A list of point-by-point replies to the reviewers' comments is reported in the following.

**Reviewer 1**

1. ***Reviewer***: *This manuscript deals with LES simulations of a down-scaled wind turbine model, and their assessment against wind tunnel data. My main comment is that this manuscript reads more as a technical report rather than a scientific paper. Majority of the text is devoted to technical details of the LES that often are not novel and already well established in the scientific community.*

   **Authors**: The novelty of this paper is in the comparison of simulation results with scaled experimental data in the context of wake control methods. The following table reports all relevant papers that we were able to locate in the literature, dealing with the three aspects of simulation models, experimental results and wake control:

| Paper title | Publication year | Scaled or full scale | # of turbines | Experimental measurements (none, hotwire, PIV, LiDAR) | Wake control | Controller in the loop |
|---|---|---|---|---|---|---|
| **Present paper: A Large-Eddy Simulation Approach for Wind Turbine Wakes and its Verification with Wind Tunnel Measurements** | - | **Scaled** | **1** | **Yes** | **Yes** | **Yes** |
| Large-eddy simulation of atmospheric boundary layer flow through a Wind Farm Sited on Topography | 2017 | Full | 1 | Yes | No | No |
| Self-similarity and flow characteristics of vertical-axis wind turbine wakes: an LES study | 2017 | Scaled & Full | 1 | Yes | No | No |
| Evaluation of layout and atmospheric stability effects in wind farms using large-eddy simulation | 2017 | Full | 48 | No | No | No |
| Modelling of wind turbine wake using large eddy simulation | 2017 | Scaled | 1 | Yes (only power) | No | No |
| Wake flow in a wind farm during a diurnal cycle | 2016 | Full | 18 | No | No | No |
| Wind plant power optimization through yaw control using a parametric model for wake effects - a CFD simulation study | 2016 | Full | 6 | No | Yes | Yes |
| Large eddy simulations of the flow past wind turbines: actuator line and disk modeling | 2015 | Full | 1 | Yes | No | No |
| Simulation comparison of wake mitigation control strategies for a two-turbine case | 2015 | Full | 2 | No | Yes | Yes |
| Simulation of wind turbine wakes using the actuator line technique | 2015 | Scaled | 1 | Yes | No | No |
| Large-eddy simulations of the Lillgrund wind farm | 2015 | Full | 48 | Yes | No | No |
| Evaluating techniques for redirecting turbine wakes using SOWFA | 2014 | Full | 1 | No | Yes | Yes |
| A numerical study of the effects of wind direction on turbine wakes and power losses in a large wind farm | 2013 | Full | ~80 | No | No | No |
| Large eddy simulation of the wind turbine wake characteristics in the numerical wind tunnel model | 2013 | Scaled | 1 | Yes | No | No |
| A Large-Eddy Simulation of Wind-Plant Aerodynamics | 2012 | Full | 37 | Yes (only power) | No | Yes |
| Large-eddy simulation of a very large wind farm in a stable atmospheric boundary layer | 2011 | Full | 2 | Yes | No | No |
| Large-eddy simulation of atmospheric boundary layer flow through wind turbines and wind farms | 2011 | Scaled & Full | 1 | Yes | No | No |
| Large-eddy simulation of wind-turbine wakes: evaluation of turbine parametrisations | 2011 | Scaled | 1 | Yes | No | No |
| Effects of thermal stability and incoming boundary layer flow characteristics on wind-turbine wakes: a wind-tunnel study | 2010 | Scaled | 1 | Yes | No | No |
| Application of a LES technique to characterize the wake deflection of a wind turbine in yaw | 2010 | Scaled | 1 | Yes | Yes | No |

   By looking at the table, it appears that papers addressing numerical simulations for wake control do not present experimental validations, while papers that validate their numerical results with experiments do not consider wake control. We believe that this table shows that our contribution has novel aspects that deserve publication, as it presents for the first time a validation (although a partial

one, as better explained in our reply to question 2) of numerical simulations for wake manipulation. The introduction has now been modified to better explain the focus and novelty of the paper. This was also reflected in a new title, which we think better describes the actual content and contribution of the paper.

We agree with this reviewer (as well as with the second reviewer, who has a similar remark) that too much text has been devoted to technical details of LES. This issue was addressed by removing unnecessary text, whenever the same information can be found elsewhere in the literature. We have also tried to streamline the presentation, and we have greatly simplified the organization of the paper into section, sub-sections and sub-subsections. We think that this new version has a much improved readability.

2. *Reviewer: In my opinion, if the focus of the paper is proposing new capabilities for an LES solver for wind turbine wake simulations, then these potential novelties should be highlighted already from the abstract. Furthermore, for each control capability, assessment against wind tunnel data for one or few conditions might not be enough to assess the accuracy of the model. Maybe, the work can be extended by detecting flow features that were not easily and well detected through the wind tunnel experiments and now better singled out through the LES simulations.*

**Authors**: As explained in the reply to the previous question, the focus of this paper is actually to use an existing LES formulation -albeit with some small but important modifications- and demonstrate its accuracy with respect to a unique set of experimental observations. This fact has now been better explained in the revised version of the introduction.

It is true that a detailed validation of the procedures would probably require an even more extensive data set. We also agree that CFD can be used to complement an experimental data set, and can help explain complex features of the flow. We are indeed finishing another paper where we use CFD to explain certain characteristics of deflected wakes that we have observed in our wind tunnel experiments. However, any of these discussions would considerably lengthen the paper. We believe that the paper in its present form -now that it has an improved introduction- achieves a clear and focused goal: providing a first evidence on the accuracy of LES methods in predicting the detailed behavior of wakes that have been "manipulated" for control purposes. This contribution appears to be novel, as the current literature has not addressed this same problem. We believe that adding now extra material, as suggested by the reviewer, would not only lengthen the paper, but would also dilute its message.

**Reviewer 2**

1. *Reviewer: The paper is too long and there are many unnecessary details about the simulations that do not need to be there. Also, the authors missed many references in the introduction, although some of them were cited later during the discussion of the results. The authors should collapse many of the sections and instead of explaining all the details about linear solver, etc., they should just cite the appropriate references. Many of the details are not relevant to the findings in the paper.*

**Authors**: Many unnecessary details have now been removed.

2. *Reviewer: The authors conclude that the main differences in power and thrust between experiment and simulations come from the blade polars. However, it looks more like the differences are coming from turbine parametrization, such as epsilon, integral sampling, etc.*

**Authors**: We have carefully investigated the possible sources of error, including the choice of epsilon and of the sampling method. Based on our results, we believe that the differences are caused by the polars. Indeed, one can tune epsilon to match the experimental results for each value of the curtailment factor. There is, however, not a single epsilon that is able to accommodate the

investigated range of curtailments. On the other hand, keeping epsilon fixed, one can observe that the errors in power and thrust grow as the extent of power curtailment increases. In our opinion, this means that the slopes of the lift and drag coefficients with respect to the angle of attack are not calibrated well. We have updated the text to explain these facts, and better support our conclusions.

3.  ***Reviewer***: *In the introduction, when talking about ALM, the authors are missing some references. Please cite the original reference of the ALM (Sorensen and Shen 2002). There are good guidelines in the literature for choosing the appropriate tuning parameters for the ALM, but they require very fine grid resolutions (Jha et al JSEE 2014, Martinez et al WE 2017, etc). Also, the effect of nacelle and tower has been studied by many (Churchfield et al AIAA 2015, Santoni et al. WE 2017, Stevens et al, RE 2018, Yang and Sotiropoulos WE 2018, etc).*
    **Authors**: The mentioned papers were cited and included in the list of references.

4.  ***Reviewer***: *In the section Sub-grid scale model, there are no references to the work of others. The effect of SGS model and Smagorinsky coefficient on wind turbine wakes has been studied before (Sarlak et al RE 2015, Martinez et al. JRSE 2018, etc).*
    **Authors**: The work of Sarlak and Martinez has been cited and their conclusions are well in line with ours, as noted in the paper.

5.  ***Reviewer***: *The section "Numerical discretization and solution of the resulting linear systems" seems to give too much information and not all of it is necessary. This section should be cleaned up.*
    **Authors**: This part has been drastically reduced.

6.  ***Reviewer***: *In the section "Computational mesh" there is a complicated explanation about wall model and y plus. I believe this explanation is not needed and the authors can just mention that a wall model is used.*
    **Authors**: This part was shortened, to accommodate the reviewer's comment.

7.  ***Reviewer***: *Figure 2: Why is the velocity not symmetric about y=0? "Baseline simulation and parameter tuning" It is not clear how the optimal parameters were chosen. Do these values agree with the optimal values found in the literature? Please add this to the discussion*
    **Authors**: The horizontal velocity profile is slightly unsymmetrical because of the not exact uniformity of the flow in the tunnel. This is due to the 16 fans of the tunnel (in two rows of eight side-by-side fans), to stiffening transects upstream of the chamber inlet, and due to the turbulence-generating spires. These explanations have been added to the text of the modified manuscript.

8.  ***Reviewer***: *Figure 5: Please move the legend outside of the figures. It would be best to just show the span-wise profiles in this figure (y/D). If it is needed, another figure showing the wall normal direction (z/D) can be added.*
    **Authors**: The legends do not cover the curves. Moving them outside of the figures would further increase their size. For these reasons, we would like to keep the figures as they are. Similarly, the addition of a separate figure for the wall normal velocity component would only increase the length of the paper, and does not appear to us to be strictly necessary.

9.  ***Reviewer***: *Figure 6: Can you make the plots wider, such that differences are shown better?*
    **Authors**: As required, the plots were made wider. Additionally, the 100% and 92.5% cases were merged together in a single plot.

10. **Reviewer**: *"Power derating" The authors conclude that the difference in polars is causing the differences in power. It seems unlikely that the polars would change much over the small Reynolds number variations seen by the blade. It seems that these differences are more due to the turbine parametrizations (epsilon, integral sampling, etc).*

    **Authors**: The change of polars with respect to the Reynolds number is small, but not completely negligible, given the very low Reynolds regime at which these airfoils operate. However, the most important factor in the mismatch is the rate of change of lift and drag with angle of attack, which considerably affects the results when the machine is derated (and, therefore, when the angle of attack at the blade sections changes). As explained when answering to question 2, tuning other parameters can only match one operating condition, but not all derating conditions (unless one is willing to accept different values of epsilon for different deratings, which does not seem to be a satisfactory solution to us). The text has been improved to better explain this point.

In addition to the changes explained above, we have taken this opportunity to modify the text where necessary and improve the quality of some figures. A revised version of the manuscript is attached to the present reply. The main changes to the text are highlighted in blue. Please notice however that many other changes were made to the text in addition to the blue highlights, including the elimination of substantial parts of the previous descriptions of the LES solver. Thanks to the reviewers' input, we believe that the paper has been greatly improved, and results in a clearer message and more informative and easier reading than in the original submission.

Best regards.
The authors

---

## Author Response (AR2)

The corrections to the bibliography have been implemented.

Best regards

The Authors